# How strongly does diet variation explain variation in isotope values of animal consumers?

Jean-François Arnoldi[1], Jenny Rose Bortoluzzi[2], Hugh Rowland[7], Chris Harrod[3,4,5], Andrew C. Parnell[6], Nicholas Payne[2], Ian Donohue[2], Andrew L. Jackson[2]*

1 Theoretical and Experimental Ecology Station, CNRS Moulis, Moulis, France, 2 Department of Zoology, School of Natural Sciences, Trinity College Dublin, Dublin, Ireland, 3 Instituto de Ciencias Naturales Alexander von Humboldt, Universidad de Antofagasta, Antofagasta, Chile, 4 Instituto Antofagasta, Universidad de Antofagasta, Antofagasta, Chile, 5 Millennium Nucleus of Austral Invasive Salmonids (INVASAL), Concepciòn, Chile, 6 Insight Centre for Data Analytics, Hamilton Institute, Maynooth University, Kildare, Ireland, 7 Independent Researcher, Dublin, Ireland

☯ These authors contributed equally to this work.
* jacksoan@tcd.ie

**Data Availability Statement:** The data are simulated data and Python and R code available on Github https://github.com/AndrewLJackson/isoVarEcol.

## Abstract

Analysis of stable isotopes in consumers is used commonly to study their ecological and/or environmental niche. There is, however, considerable debate regarding how isotopic values relate to diet and how other sources of variation confound this link, which can undermine the utility. From the analysis of a simple, but general, model of isotopic incorporation in consumer organisms, we examine the relationship between isotopic variance among individuals, and diet variability within a consumer population. We show that variance in consumer isotope values is directly proportional to variation in diet (through Simpson indices), to the number of isotopically distinct food sources in the diet, and to the baseline variation within and among the isotope values of the food sources. Additionally, when considering temporal diet variation within a consumer we identify the interplay between diet turnover rates and tissue turnover rates that controls the sensitivity of stable isotopes to detect diet variation. Our work demonstrates that variation in the stable isotope values of consumers reflect variation in their diet. This relationship, however, can be confounded with other factors to the extent that they may mask the signal coming from diet. We show how simple quantitative corrections can recover a direct 1:1 correlation in some situations, and in others we can adjust our interpretation in light of the new understanding arising from our models. Our framework provides guidance for the design and analysis of empirical studies where the goal is to infer niche width from stable isotope data.

## Introduction

Understanding the factors driving variation in niche use across levels of biological organisation is central to ecological concepts such as resource use, geographic diversity, community

**Funding:** This study was financially supported by an Irish Research Council Laureate Award (IRCLA/2017/186) received by ALJ in support of ALJ, JRB and J-FA. This study was also financially supported by ANID – Millennium Science Initiative Program in the form of an award (NCN2021-056) received by CH. The funders had no role in study design, data collection and analysis, decision to publish, or preparation of the manuscript.

**Competing interests:** The authors have declared that no competing interests exist.

composition and structure, and the effects that events such as competition, global change, extinction or invasive species will have for conservation biology and applied ecology [1]. Of particular interest has been understanding how niches vary over scales from individuals [2–5] through to populations [6] and communities [5] over time and space. It is often difficult to directly observe and quantify an organism's niche and so indirect methods using naturally occurring biotracers such as stable isotope or fatty acid analyses have become commonplace both to provide insight in to the trophic (bionomic) and/or environmental (scenopoetic) niche [7]. These methods rely on natural variation in tracer over time and space to make a direct link to variation in consumers' use of the resources containing that tracer [7, 27].

The use of stable isotopes to characterize community structure and niche width has become increasingly common since first proposed in 1978 [7–10]. Particular ratios of heavy to light stable isotopes are found in the tissue of consumers after accumulating through the food web via the process of assimilation (either through primary production or consumption) and metabolic incorporation. They have, therefore, been proposed as a way to quantify the niche space of individuals or populations, resulting in the concept of "isotopic niche" [9, 11], via the "you are what you eat" premise [12]. The isotopic niche—that is, the area occupied by a group of measurements in isotopic space—is effectively a measure of variation among and within individuals in isotopic values [7]. While the isotopic niche is often equated to the dietary or environmental niche [13], it has received justifiable criticism. In addition to diet, environmental, behavioural and physiological factors also affect isotopic variance among individuals [10, 11, 14, 15]. It has been argued that trophic and isotopic niche spaces must be kept conceptually distinct and the relationship between the isotopic niche and niche overlap among individuals is complicated and unknown [16]. They also note that individual variation at the physiological level impact consumer isotope values. For example, the turnover rates of elemental pools affect stable isotope ratios in consumers, which vary depending on the tissue and on the species and potentially the individual animal, translate into an isotopic niche more or less closely correlated to the ecological niche of an individual, population or community.

Here, we explore the general biomass incorporation model that forms the basis of our understanding of stable isotope ecology [17–19]. We ask in a quantifiable manner "what drives observable variation in stable isotope values of consumers?" We use a series of vector algebra methodologies to determine how variation in each of the main processes underlying elemental incorporation are expected to result in variation in the tissues of a consumer. Variation in these processes might take the form of purely random noise, or they may be more structured such as in the case of diets that might vary through time or space in a correlated manner. Ultimately, the key objective is to determine how strong the signal of diet variation is within the emergent variation of stable isotope values in the consumer(s). We show concretely that variation in the isotope values of the food sources (baseline variation) is a major factor with nuanced manifest effects on consumer isotope variation to the extent that under certain extreme situations, all variation in consumers might be caused by baseline variation. We find that variation in diet expressed in terms of Simpson index rather than simple variance is the appropriate quantitative link to variation in consumer stable isotope values. We discuss implications for empirical study-design, statistical analyses and the assumptions made in this theoretical model.

## Materials and methods

We start by considering a single organism and its isotopic ratio $y$, such as the ratio of rare heavy nitrogen $^{15}N$ over the lighter common form $^{14}N$ measured in a sample compared with a standard [17]. The consumer isotopic content reflects the assimilation of $S$ different sources,

each source having a slightly different stable isotope ratio. For a heterotrophic animal, these sources would represent discrete food or prey items that are ingested and assimilated to form various tissues of the organism.

Although biologically complicated, the entire assimilation process can be abstracted as a continuous flux of elements comprising their isotopes into an organism from its food sources, with the organism constructing its tissues from this flux and continually replacing the elements of its own tissues. Whereas ecologically we are interested in quantifying diet as the proportion of biomass $p_i$ assimilated from a given food source $i$, from an isotope assimilation perspective we need to consider how much of a given element forms the flux into the organism $f_i$ (i.e. the concept of "concentration dependence" in [20]). These two quantities, $(p_i)$ and $(f_i)$, which both define points on the simplex of dimension $S$ (vectors whose $S$ entries are positive and collectively sum to one), are related via the elemental concentration in a given food source $c_i$ such that

$$f_i = \frac{c_i p_i}{\sum_{j=1}^{S} c_j p_j}.$$ (1)

In the case where all $c_i$ are equal or approximately equal, the biomass contribution proportions are the same as the isotope contribution proportions $p_i = f_i$.

In the following we focus on $\boldsymbol{f} = (f_i)$ the vector of relative fluxes of the element in question (typically carbon or nitrogen) and explore its relationship with the consumer isotopic ratio $y$.

## Model

In the ideal case where our organism is at perfect isotopic equilibrium with its food sources, we have a simple model for the consumer isotopic ratio $y$:

$$y = \langle \boldsymbol{f}, \boldsymbol{x} \rangle + \psi$$ (2)

here $\boldsymbol{x}$ is the vector of isotope ratios of the $S$ food sources, combined with diet by the scalar product $\langle \boldsymbol{f}, \boldsymbol{x} \rangle = \sum_i f_i x_i$, which amounts to a weighted mean across food sources (recall that $\boldsymbol{f}$ is a point on the simplex of dimension $S$, that is a vector whose $S$ entries are positive and collectively sum to one). The term $\psi$ is an additive trophic discrimination factor representing the metabolic phenomenon of incorporation which takes values specific to the tissue sampled in the consumer. This equation is often articulated as "you are what you eat plus a few per mille" [17].

This model expresses, in the simplest possible way, a formal relationship between the diet and the isotopic content of a consumer.

Here, given a population sample of $y$, so a set of isotopic measurements $(y(w))_{w=1,...,W}$ of $W$ individual consumers, we want to know how the variance of this sample relates to the variation in diet across consumers. We could try to derive this relationship based on the simple model (Eq 2). The assumptions on which it is based, however, are not quite reasonable since organisms are unlikely to be at isotopic equilibrium with their food sources –this would require a constant diet for any given individual and fixed isotopic ratios of all resources in time and across the landscape (see Box 1). To refine our model, we can first recognize that the integrative process inherent to assimilation means that fluctuations in the diet and in isotope values of the food sources are averaged over a time window defined by the rate of tissue turnover $r$. The simplest, proper way to do so is to weight the isotopic or diet values at a past time $t - \tau$ by a factor $d\mu(\tau) = re^{-r\tau}d\tau$ (note that $\int_0^\infty d\mu(\tau) = 1$). A maximal weight is given to present values while values at times earlier than approximately $t - 5/r$ (which is a somewhat arbitrary value, and is numerically close to and similar in concept to the value $t - 2\log(2)/\log(r)$ suggested in a

**Box 1** Consider a given food source $i$, distributed across the landscape $\Omega \subset \mathbb{R}^d$ where the consumer population live and is sampled. For a given element, the isotopic ratio $x_i$ of the food source is not constant, but varies across the landscape. Denote spatial coordinates by $\vec{z} \in \Omega$ and $x_i(\vec{z})$ the spatial distribution of isotopic ratio of food source $i$. Prior to the sampling event a given consumer $w$ has moved across the landscape along a particular trajectory

$$\vec{z}_w : t \mapsto \vec{z}_w(t) \in \Omega.$$

Therefore, from the perspective of this consumer, the relevant isotopic signal of food source $i$ is, at the sampling time $t_w$

$$x_{i,\mu}(t_w, w) = \int_0^\infty x_i(\vec{z}_w(t_w - \tau)) d\mu(\tau) = \bar{x}_i^\Omega + \Delta x_{i,\mu}(t_w, w)$$

here $\bar{x}_i^\Omega$ is the spatial average of $x_i(\vec{z})$ and $\Delta x_i(\vec{z}) = x_i(\vec{z}) - \bar{x}_i^\Omega$ the deviation from the mean at each point in $\Omega$. Thus $\Delta x_{i,\mu}(t_w, w)$ is the smoothed version of $\Delta x_i$ along the consumer's trajectory, at time $t_w$. To make this notion empirically accessible we need to make some simplifying assumptions. If we can assume that $\bar{\Delta x}_\mu \approx 0$, so that consumers, overall, sample evenly the landscape, we get that

$$\bar{x}_\mu \approx \bar{x}_\mu^\Omega \Leftrightarrow P\bar{x}_\mu \approx P\bar{x}_\mu^\Omega; \text{ and } \operatorname{var}(x_\mu) \approx \operatorname{var}(\Delta x_\mu).$$

The vector of spatial means could be inferred in practice but not $\operatorname{var}(\Delta x_{i,\mu})$. We can still make some useful estimates based on knowledge of the spatial distribution of isotopic ratios in the food sources. If we know the spatial variance $\operatorname{var}_\Omega(x_i)$ of isotopic values, we can expect an upper bound

$$\operatorname{var}(\Delta x_{i,\mu}) \leq \operatorname{var}_\Omega(x_i)$$

if we have some notion of distance of variation of isotopic fluctuations $d_i$, and a notion of speed $v$ of movement across the landscape of consumers, then given the tissue turnover rate $r$ the effective distance over which the individual would average any given fluctuation is

$$d_\mu = v/r$$

If $d_i/d_\mu \ll 1$ fluctuations are completely averaged out by consumer assimilation and so $\operatorname{var}(\Delta x_{i,\mu}) \approx 0$. On the other hand if $d_i/d_\mu$ is very large, the fluctuations are not averaged out at all, so that $\operatorname{var}(\Delta x_{i,\mu})$ should be expected to approach the spatial variance of isotopic fluctuations $\operatorname{var}_\Omega(x_i)$. We can interpolate between these two extreme cases as

$$\operatorname{var}(\Delta x_{i,\mu}) \approx \frac{d_i/d_\mu}{1 + d_i/d_\mu} \operatorname{var}_\Omega(x_{i,\mu})$$

All in all we finally get that the elements of baseline isotopic variation are

$$||P\bar{x}_\mu||^2 \approx ||P\bar{x}_\mu^\Omega||^2$$

and

$$||\text{std}(\boldsymbol{x}_\mu)||^2 \approx ||D_\mu \text{std}_\Omega(\boldsymbol{x})||^2; \ D_\mu^2 = \text{diag}\left(\frac{d_i/d_\mu}{1 + d_i/d_\mu}\right)$$

related formulation [24], being too far in the past, essentially do not contribute to present observations. In this formulation we assume that turnover rates are the same for all individuals, but of course this is another potential source of variation that has been identified as important in some systems [26] and may be worthy of further exploration for specific cases outside our more general investigation of the effects of diet variation.

Applying this idea to model the isotopic content in an organism $y$ at time $t$ gives a more sophisticated expression

$$y(t) = \int_0^\infty \langle \boldsymbol{f}(t - \tau), \boldsymbol{x}(t - \tau) \rangle d\mu(\tau) + \psi \tag{3}$$

This expression is more realistic than model (Eq 2) but difficult to work with, as the integral over past times entangles the scalar product between diet and isotopic sources. Yet we can recognise that this integral is a weighted mean through time, with the weighting coming from $d\mu(\tau)$ and the average achieved by summation by the integral. By definition, the mean of the product of uncorrelated variables is the product of their respective means. Thus, as long as diet temporal fluctuations $f_i(t)$ are not too strongly correlated with fluctuations of isotopic values $x_i(t)$, we can simplify the above and recover an expression similar to the most simple model such that we consider a mean isotopic value in the consumer as a function of mean diet and mean isotopic value of the food sources (Eq 2)

$$y(t) \approx \langle \boldsymbol{f}_\mu(t), \boldsymbol{x}_\mu(t) \rangle + \psi \tag{4}$$

This assumption of independence more than likely holds in most situations since it unlikely that consumers are changing their diets in response to fluctuations in the isotopic values of the food sources, i.e. it would be odd if consumers were actively changing diets to feed on the isotopically heaviest food source. It is possible though that some seasonal fluctuations in isotopic values might coincide with diet fluctuations and introduce some correlation indirectly, but this would require some unlikely alignment of dynamic processes.

Non-equilibrium dynamics, which result in temporal variations of isotopic values within an individual, are taken into account by a change of interpretation of diet and isotopic values. One should see them not as snapshots at a given time but time-averaged over a historic temporal window, determined by the rate of tissue turnover. This simple idea is formalized by denoting, for any general function of time $h(t)$, its smoothed version (according to the weight $\mu$), $h_\mu(t) = \int_0^\infty h(t - \tau) d\mu(\tau)$. Thus, we should rephrase the simply worded isotope model to be "you are what you *have been eating* plus a few per mille"; where your diet might be changing over time either stochastically or through preference and/or the isotopic content of your food might also be changing either stochastically or fluctuating perhaps according to seasonal cycles [19].

## Analytical predictions

Based on the above reasoning, we model the isotopic ratio of an individual $w$, sampled at $t_w$, as

$$y(w) = \langle \boldsymbol{f}_\mu(t_w, w), \boldsymbol{x}_\mu(t_w, w) \rangle + \psi(w) \tag{5}$$

In what follows we will make use of the linear structure of this expression to determine a precise relationship between isotopic variance across samples in a population and diet variation. We will treat the vector-valued variables $\boldsymbol{x}_\mu$, and $\boldsymbol{f}_\mu$ and the scalar variable $\psi$ as *independent random variables*, whose realizations $\boldsymbol{f}_\mu(t_w, w)$, $\boldsymbol{x}_\mu(t_w, w)$ and $\psi(w)$ correspond to the sampling of an individual $w$. We focus on basic summary statistics, using the following notations: let $\bar{h}$ be the empirical population mean of some sample set $\{h(w); w = 1, \ldots, W\}$. Let $\mathrm{var}(h)$ be its the empirical variance and $\mathrm{std}(h)$ its standard deviation. When dealing with diet or source terms, which are organized as vectors with $S$ elements ($S$ being the number of potential food sources) we will distinguish population statistics of a given entry from the statistics across entries. For a sample of vectors $\{\boldsymbol{h}(w)\}$, $\bar{\boldsymbol{h}}$ is the vector of empirical means, $\mathrm{var}(\boldsymbol{h})$ the vector of variances, and $\mathrm{std}(\boldsymbol{h})$ the vector of standard deviations. By contrast, the mean across elements is represented algebraically as $\langle \boldsymbol{h}, \mathbf{1} \rangle / S$ where $\mathbf{1}$ is a vector whose elements all equal to 1. The variance across elements is computed after subtracting the mean from all elements. This operation defines the projector $P$ as

$$P\boldsymbol{h} = \boldsymbol{h} - \frac{\langle \boldsymbol{h}, \mathbf{1} \rangle}{S} \mathbf{1} \tag{6}$$

Then, $||P\boldsymbol{h}||^2/S$, where $||\cdot||$ denotes the euclidean norm of vectors, is the variance across elements of $\boldsymbol{h}$. For positive vectors that sum to one (such as diet vectors $\boldsymbol{f}$) we denote their Simpson index ($\Lambda$), a reciprocal measure of diversity, and its bounds as

$$1/S \leq \Lambda(\boldsymbol{f}) = ||\boldsymbol{f}||^2 \leq 1. \tag{7}$$

With these notations we can state our first claim, a general inequality between diet diversity and isotopic variance once expressed relatively to a baseline. Let

$$0 \leq \phi = \frac{||P\bar{\boldsymbol{x}}_\mu||^2}{||\mathrm{std}(\boldsymbol{x}_\mu)||^2 + ||P\bar{\boldsymbol{x}}_\mu||^2} \leq 1 \tag{8}$$

denote the relative contribution of inter-specific isotopic variation across food sources, to the overall isotopic variations present in the system. A situation where each food source has a fixed, unvarying, isotopic signal corresponds to $\phi = 1$. By contrast, $\phi = 0$ corresponds to the case where the variation in the isotopic landscape only comes from intra-specific variation (i.e. the mean isotope values of the food sources are identical). Then

$$\boxed{0 \leq \frac{\mathrm{var}(y) - \mathrm{var}(\psi)}{\mathrm{baseline}} \leq \frac{S}{S - \phi}\left(\bar{\Lambda}(\boldsymbol{f}_\mu) - \phi\,\Lambda(\bar{\boldsymbol{f}}_\mu)\right) \leq 1} \tag{9}$$

where

$$\mathrm{baseline} = ||\mathrm{std}(\boldsymbol{x}_\mu)||^2 + \frac{S - 1}{S}||P\bar{\boldsymbol{x}}_\mu||^2 \tag{10}$$

is the sum of all intraspecific isotopic variances of food sources, plus $(S - 1)$ times the variance across food sources. This sets the absolute baseline isotopic variation, a fraction of which is reflected in the isotopic variance of the consumer population. The more food sources (the

larger $S$ is), the larger this baseline is expected to be. This fundamental term is explained in more detail in Box 1. Moreover,

- var($\psi$) is an additive term that arises from variations in fractionation and other additive sources of variance, such as measurement error of $y$.

- $\bar{\Lambda}(\boldsymbol{f}_\mu)$ is the mean Simpson index of diet. That is, Simpson index is calculated for each individual's diet and then the mean is taken over the population. This value is maximised when diets fall at the corners of the simplex (all individuals specialize on a single resource) and minimised when all diets are on the centre of the simplex (all individuals are generalists).

- $\Lambda(\bar{\boldsymbol{f}}_\mu)$ is the Simpson index of mean diet. That is, mean diet over all individuals is evaluated before computing its Simpson index. This measures the degree of specialism of the population as a whole, it is maximized if all individuals feed from the same single resource.

- The difference $\bar{\Lambda}(\boldsymbol{f}_\mu) - \Lambda(\bar{\boldsymbol{f}}_\mu)$ coincides with total diet variance $||\text{std}(\boldsymbol{f}_\mu)||^2$, since it is effectively a measure of the deviances around a mean.

Eq 9 is a rigorous statement (under the assumption of statistical independence between realizations of diet and food source isotopic signals, see discussion) but not necessarily the most informative to relate isotopic variance to diet variation. A complementary approach, at the cost of generality, is to look for a generic relationship, reflecting the expected outcome of typical scenarios for the values taken by the statistics (mean and covariance) of diet and food source isotopic signals. This generic relationship can be deduced by using basic results of random geometry, which we explain in S1 File. But in the end, it basically amounts to normalizing the baseline by a factor $S$, the number of food sources (this normalization could be guessed by looking at the way systems with larger food source dimensionality behave in Fig 1). Our second claim is the proposition of such a generic relationship, that is essentially 1-to-1 in the absence of systematic correlations and/or highly anisotropic diet distributions. This approximation goes as follows, if

$$\phi^* = \frac{||P\bar{\boldsymbol{x}}_\mu||^2}{||P\bar{\boldsymbol{x}}_\mu||^2 + \frac{S-1}{S}||\text{std}(\boldsymbol{x}_\mu)||^2} \tag{11}$$

denotes the relative contribution of inter-specific isotopic variation across food sources (note the subtle difference with $\phi$ as defined previously), then

$$\boxed{\frac{\text{var}(y) - \text{var}(\psi)}{\text{baseline}^*} \approx \frac{S}{S - \phi^*}\left(\bar{\Lambda}(\boldsymbol{f}_\mu) - \phi^*\,\Lambda(\bar{\boldsymbol{f}}_\mu)\right)} \tag{12}$$

where the baseline is now taken to be

$$\text{baseline}^* = \frac{1}{S}||\text{std}(\boldsymbol{x}_\mu)||^2 + \frac{1}{S}||P\bar{\boldsymbol{x}}_\mu||^2 \tag{13}$$

that is, the mean intraspecific isotopic variance of food sources, plus the variance across food sources. It is the effective baseline isotopic variation, a fraction of which is reflected in the isotopic variance of the consumer population (but now this fraction could exceed one). It is essentially the absolute baseline of claim (Eq 9) divided by the number of food sources.

We insist that this last result is an approximation, seen as a null model from which specific examples could deviate depending on the precise relationship between the statistics of source terms and diets. In fact, inspection of the derivation shows that deviations from this

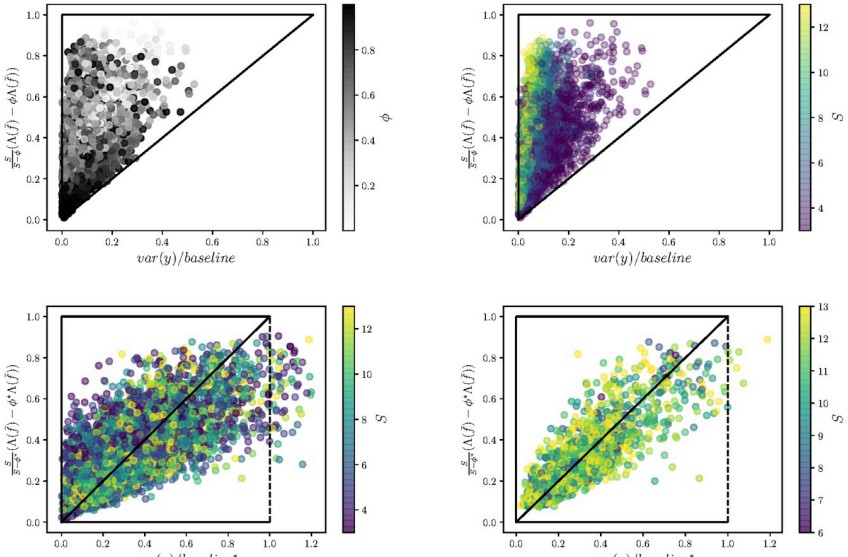

**Fig 1. Numerical test of the analytical predictions.** Top row tests the boundaries defined by the inequality expressed in Eq (9). Each point corresponds to a simulated population of 100 individual consumers feeding on $S$ food sources, and whose isotopic ratio follows model Eq (4). In the left panel the grey scale indicates the value of $\phi$ which measures the contribution of intraspecific isotopic variance in the overall variation present across all food sources. On the right we color points by the number of food sources. The bottom row tests the approximation Eq (12), which essentially amounts to normalizing the baseline of the top row by $S$. On the right we only consider diet distributions that are not too anisotropic (Simpson index of the eigenvalues of the covariance matrix of diet is less than 1/4).

expectation require the diet distribution to be anisotropic, with diet variations taking place along favoured directions of the simplex.

## Simulations

We simulated the sampling of isotopic values of some hypothetical element across a population that would perfectly follow model Eq (4). We therefore needed to generate samples of diet vectors $f_\mu(w)$ and samples of isotopic source ratios $x_\mu(w)$ and take their scalar product to get the consumer's isotopic sample $y(w)$. In simulations, we do not explicitly model the effect of time integration (encoded by the weighted mean with weight $\mu(\tau)$). We also drop the fractionation term $\psi$, as it only acts as an additive term in the variance. We simply generated a population of vectors on the simplex $\{f(w)\}$ and a corresponding population of positive vectors $\{x(w)\}$. To do so, given a number of food sources $S$ drawn between 3 and 13, we drew $W = 100$ individual samples, seen as 100 individual consumers. To generate their diet we used a multivariate Dirichelet distribution, so $f(w) \sim \text{Dir}(\boldsymbol{\alpha})$. We decomposed the parameter vector as $\boldsymbol{\alpha} = \alpha_0 \boldsymbol{n}$, with $1/\alpha_0$ controlling the variance of the distribution and $\boldsymbol{n}$ is its mean. We drew $\alpha_0$ uniformly between 1 and 10. To evenly sample the means $\boldsymbol{n}$ we first drew a vector $\boldsymbol{X} \sim \mathcal{N}(0, 1)$ in a multivariate standard normal distribution. Once normalized this gives a uniform sampling of the sphere, so points whose squared elements sum to one. We then took $\boldsymbol{n}$ to be the the vector whose elements are the square of the elements of $\boldsymbol{X}/||\boldsymbol{X}||$. For the source terms $x(w)$, we first drew their means $\mu_i$ uniformly between 0 and 10. We then drew a parameter $\sigma$ uniformly between 0 and 1 to set the level of intraspecific variance. Then $x_i(w) = \mu_i(1 + \sigma\epsilon_i(w))$ with $\epsilon_i(w)$ drawn uniformly between -1 and 1. The isotopic content of the 100 individuals is then given by $y(w) = \langle f(w), x(w) \rangle$. Taking the population variance gives one data point. We reiterated this

procedure 10000 times. The anisotropy of the diet distribution was recorded as the Simpson index of the set of eigenvalues of its covariance matrix.

## Results and discussion

Based on a general model for the processes of dietary incorporation of stable isotopes (Eq 3) we revealed formal relationships that exists between the empirical isotopic variance across consumers, and their dietary variation. We derived two complementary results. The first is a general inequality, showing how a specific measure of dietary variation sets the upper bound for isotopic variance Eq 9. The second result is a generic expectation Eq 12, stating that, once properly rescaled by the number of food sources, the isotopic variance is expected to coincide with a specific measure of dietary variation (essentially the same measure as the one that sets the upper bound).

Our results, illustrated by simulated data modelling a wide spectrum of situations (Fig 1), clarify two major points. This first should not be surprising: it is not the absolute observed isotopic variance across consumers that is informative, but the fraction that it represents of the baseline isotopic variation across food sources present in the system. Less obvious is what this baseline is. Here we provide precise expressions. For the generic claim in particular, it is the variance across food-sources plus the mean variance *within* food-sources (this is Eq 13 but see Eq 10 for the general upper bound). The importance of baseline variance has been highlighted numerous times [4, 13, 15, 21] but to our knowledge our model is the first to clearly point to a mechanistically argued and demonstrably accurate mathematical correction to enable comparisons between study systems where baselines differ.

The second –arguably more novel– result precisely clarifies which measure of dietary variation is expressed by isotopic variance. It is a linear combination of two Simpson diversity indices $\bar{\Lambda}(f)$ and $\Lambda(\bar{f})$. The former is the mean, across consumers, of the Simpson index of their respective diets. The latter is the Simpson index of the mean diet. How the combination should be made depends on how much isotopic variance exists *within* food-sources.

When there is no such intra-specific variance, so each food source has a fixed isotopic signal the proper combination of indices is their difference, which turns out to be the sum of the variance of dietary proportions. Focusing on the generic case Eq (12) for simplicity

$$\frac{\text{var}(y) - \text{var}(\psi)}{\text{baseline}^*} \approx \frac{S}{S-1}\left(\bar{\Lambda}(f_\mu) - \Lambda(\bar{f}_\mu)\right) = \frac{S}{S-1}\sum_i \text{var}(f_{\mu,i}) \tag{14}$$

In this case, isotopic variance is directly proportional (up to the additive term) to total diet variance. On the other extreme, if all the baseline variation is due to isotopic variation within food sources then only the first index plays a role and we get that

$$\frac{\text{var}(y) - \text{var}(\psi)}{\text{baseline}^*} \approx \bar{\Lambda}(f_\mu) \tag{15}$$

so now isotopic variance is proportional to the average of the diet Simpson diversity index. When there is no variation between the means of the food sources, the variation among consumer isotope values reflects the average diversity of their diet. If this diversity is minimal, so each consumer effectively eats a single resource, then the isotopic variance is essentially that of a single food source. On the other hand, if the diet is maximally diverse, so all consumers evenly eat from all $S$ food sources, the isotopic variance of consumers is minimal. Consumers sample isotopes from $S$ sources, so that their isotopic signal is the mean source isotopic value up to a standard error. This standard error is precisely the Simpson diversity index.

Situations will arise where the isotopic values of the food sources are not consistent between locations where, or at times when, the consumers were sampled. The results shown in Fig 1 illustrates that it is entirely possible to have situations where variation in consumer isotope values do not coincide with diet variation. By taking an approximation Eq (12) and dividing consumer isotope value by baseline* which comprises an estimate of the mean variance within sources and the variance among sources, it is possible to recover an approximate 1:1 relationship between corrected consumer isotope variance and variation in diet. There exists some error around this relationship, driven primarily by anisotropy in the diets (compare left bottom row Fig 1 which includes all simulations, and bottom right which restricts simulations to less anisotropy).

Our results confirm those of previous studies [4, 15] that baseline variation may confound the results by increasing consumer variation where the sources are more disparate, and vice versa. Clearly, in this scenario, the naive approach where one makes the simple assumption that variation in isotopes says something about diet variation has poor predictive power. Where possible, the source isotope data should be sampled appropriately [4, 22, 23] and the baseline computed in order to express the observed isotopic variance relatively to the baseline. There may be situations where data on the source variation is not possible to obtain and, in those situations, one must make an implicit assumption that the differences in the variance in sources between consumer populations is negligible.

While we frame our model in the context of variation among individuals, it provides equal insight into how variation in diet over time and/or space might manifest as variation with in a consumer. We show that as expected from the process of elemental incorporation, tissue turnover rates play a key role in determining how sensitive the consumer tissue will be to changes in diet [17, 24]; a process that has previously been linked with the relative body masses of consumers and their prey [25]. While it is intuitive to state that consumers are likely never at equilibrium with their diets [11], we have shown that one can shift the interpretation such that consumer isotope values do represent an historic equilibrium averaged over a temporal window defined by their tissue turnover rate. The discrete foraging process of most consumers means that they encounter and ultimately assimilate packets of elements which vary over time and/or space according to either diet and/or variation in the baseline isotope values of their foods. We consider this process by imagining consumers sampling along a trajectory of possible food sources defined by a spatial or temporal field of possibilities (Box 1). Intuitively, the faster consumers move around sampling the distribution of food sources ($v$) and the smaller the tissue turnover rate ($r$) relative to the rate of change of isotope values of the food sources ($d_i$) then fluctuations in food get smoothed out completely by assimilation. Conversely, if tissue turnover rates are relatively high, then the signal from the full distribution of foods encountered (arising from either intra- or inter-specific variation) becomes more evident in the consumer(s). It is this very process that allows us to identify how important baseline variation is and to tease apart the effect of intra- and inter-specific variation via the parameter $\phi$ (Eq 12). At one extreme, there is no variation within food sources ($\phi = 1$) and so the prediction collapses to a simple concept where consumer isotopic variance is directly proportional to diet variance. At the other extreme ($\phi = 0$) the only possible source of consumer variance is the mean Simpson index of diet. Now, while there is no isotopic difference among the means of the food sources, differences in diet among consumers means they encounter different variances as they sample the food sources according to their diets and so some food sources push more variation into the consumers compared with others.

One of the powerful aspects of this modelling approach is that, in making some simplifying assumptions, we can gain clear insights in the main drivers of variation in consumer stable isotope data. The simulations inherently relax some of these assumptions and allow us to observe

departures from this expectation; for example, in the case of anisotropic diets, driving deviation from the prediction in Fig 1 (compare left and right of bottom row). Clearly, in real systems, there are additional sources of variation and processes that might affect the behaviour of the models. Among these, human and machine errors can be controlled through calibration and standardisation of methods. These will inflate variation at either or both the consumer side or food source side and should do so in an additive sense on var($y$) and the baseline term respectively.

From the beginning, we made the assumption that both trophic discrimination factors are constant additive effects and therefore were able to ignore their effect on variance. Additionally we assumed that tissue turnover rates are constant between individuals in a population. We know that these effects can vary both within and among individuals and, along with discriminatory metabolic routing of elements, can be an additional source of variation. In ignoring this effect, we are taking the view that the effects are small relative to the other sources of variation in our model. Specific empirical systems could point to situations where this is not true, e.g. in fasting animals or during gestation, and our advice here would follow the general advice of cautioning that, without detailed knowledge of those systems and specific models to describe them, stable isotopes may not be suitable for providing insights on differences in their diet [11, 26].

Our derivations rely on the assumption of statistical independence between variations in diet and intraspecific isotopic variation of food sources. Relaxing this assumption is not a big issue if the later variation is small compared to the isotopic variance across food sources (when $\phi$ is close to one in our theory). When this is not the case, i.e. when a the isotopic signal of a given food source substantially varies in the system (thus when $\phi$ close to 0 in our theory), we can imagine specific scenarios where changes in diet are compensated by the isotopic variation of the food source, thus masking, in a way that we cannot easily untangle, the true diet variation. This limitation is a very general one, an inherent limit of isotope analysis.

## Conclusion

Our mathematical analysis of a general model of isotope incorporation provides a quantitative answer to the question of whether diet variation can be inferred by analysis of consumer isotope variation. It shows that with certain conditions and assumptions there is indeed such a detectable signal as proposed [15, 27]. However, the concerns raised previously [4, 16] point to nuances that must be considered either quantitatively using the baseline correction we propose here, or qualitatively by assuming that these sources of discrepancy are negligible for a given analysis. Specifically, it is possible to predict a 1:1 relationship between the two so long as one can either quantitatively account for baseline variation or satisfy oneself that its effect would be marginal. When considering a time series of isotope values from single individuals or indeed populations, an important constraint on the signal is the relative turnover rates of consumer tissues and diet which will control how the former tracks the latter. Even if the rates are such that they depart from our analytical predictions, we nonetheless expect to see a semi-quantitative relationship between the two, meaning that semi-quantitative comparisons are still possible. Our models give much quantitative confidence to many assumptions made through logically argued theoretical frameworks, make explicit the limitations of the applicability, and provide guidance for empirical studies going forward. With the ready access to statistical tools (such as [9, 28]) users of the stable isotope method and these analytical packages must satisfy themselves that the insights from our quantitative model apply to their system. There remains a considerable amount of work to update these statistical models in light of these new findings.

## Supporting information

**S1 File. Additional mathematics.** Derivations of the general and generic predictions.
(PDF)

## Author Contributions

**Conceptualization:** Jean-François Arnoldi, Jenny Rose Bortoluzzi, Chris Harrod, Andrew C. Parnell, Andrew L. Jackson.

**Formal analysis:** Jean-François Arnoldi, Jenny Rose Bortoluzzi, Hugh Rowland, Andrew L. Jackson.

**Funding acquisition:** Andrew L. Jackson.

**Investigation:** Jean-François Arnoldi, Jenny Rose Bortoluzzi, Hugh Rowland, Andrew L. Jackson.

**Methodology:** Jean-François Arnoldi, Jenny Rose Bortoluzzi, Andrew L. Jackson.

**Project administration:** Andrew L. Jackson.

**Resources:** Nicholas Payne, Andrew L. Jackson.

**Software:** Jean-François Arnoldi, Hugh Rowland, Andrew L. Jackson.

**Supervision:** Andrew C. Parnell, Nicholas Payne, Ian Donohue, Andrew L. Jackson.

**Writing – original draft:** Jean-François Arnoldi, Jenny Rose Bortoluzzi, Andrew L. Jackson.

**Writing – review & editing:** Jean-François Arnoldi, Jenny Rose Bortoluzzi, Chris Harrod, Andrew C. Parnell, Nicholas Payne, Ian Donohue, Andrew L. Jackson.

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
