## [Decision Letter · Decision Letter 0]

10 Oct 2023

PONE-D-23-24341Identifying the limits where variation in consumer stable isotope values reflect variation in dietPLOS ONE

Dear Dr. Jackson,

Thank you for submitting your manuscript to PLOS ONE. After careful consideration, we feel that it has merit but does not fully meet PLOS ONE’s publication criteria as it currently stands. Therefore, we invite you to submit a revised version of the manuscript that addresses the points raised during the review process.

ACADEMIC EDITOR: Thank you for submitting your work to Plos One. I apologize for the delay in the review process. Several reviewers dropped out after agreeing to review the paper. I have been cognizant of the maths presented in the paper, and I attempted to target reviewers to evaluate the math as suggested in your cover letter. However, given the delays; I have decided to move forward based on the judgement of the reviewer and my review of the paper. After revision, I will likely seek an additional reviewer. That stated, I find the math quite well presented. Integrals always demand a lot of work on the part of the reader. I recommend following Reviewer 1's advice with regard to potentially develop an example. Further, Reviewer 1 makes salient points that should be addressed to meet Plos One publication criteria and improve the manuscript (e.g., re: assumptions and implications for empirical studies).==============================

We look forward to receiving your revised manuscript.

Kind regards,

Jacob Freeman

Academic Editor

PLOS ONE

Journal Requirements:

 "ALJ, JRB and J-FA were funded by an Irish Research Council Laureate award IRCLA/2017/186 to ALJ."  

"ALJ, JRB and J-FA were funded by an Irish Research Council Laureate award

IRCLA/2017/186 to ALJ. All authors declare no conflict of interest."

 "ALJ, JRB and J-FA were funded by an Irish Research Council Laureate award IRCLA/2017/186 to ALJ."

Reviewers' comments:

Reviewer's Responses to Questions

**Comments to the Author**

1. Is the manuscript technically sound, and do the data support the conclusions?

Reviewer #1: Partly

2. Has the statistical analysis been performed appropriately and rigorously? 

Reviewer #1: I Don't Know

3. Have the authors made all data underlying the findings in their manuscript fully available?

Reviewer #1: Yes

4. Is the manuscript presented in an intelligible fashion and written in standard English?

Reviewer #1: Yes

5. Review Comments to the Author

Reviewer #1: See attached fileSee attached fileSee attached fileSee attached fileSee attached fileSee attached fileSee attached fileSee attached fileSee attached fileSee attached fileSee attached fileSee attached file

6. PLOS authors have the option to publish the peer review history of their article (what does this mean?). If published, this will include your full peer review and any attached files.

Reviewer #1: No

---

## [Author Response · Author response to Decision Letter 0]

6 Feb 2024

ACADEMIC EDITOR: Thank you for submitting your work to Plos One. I apologize for the delay in the review process. Several reviewers dropped out after agreeing to review the paper. I have been cognizant of the maths presented in the paper, and I attempted to target reviewers to evaluate the math as suggested in your cover letter. However, given the delays; I have decided to move forward based on the judgement of the reviewer and my review of the paper. After revision, I will likely seek an additional reviewer. That stated, I find the math quite well presented. Integrals always demand a lot of work on the part of the reader. I recommend following Reviewer 1's advice with regard to potentially develop an example. Further, Reviewer 1 makes salient points that should be addressed to meet Plos One publication criteria and improve the manuscript (e.g., re: assumptions and implications for empirical studies).

Authors: thank you editor for these considerations. We have addressed your points below where the reviewer raised them. 

Review of PONE D 23 24341 

By Arnoldi et al. 

The paper aims at determining if the variation in consumer stable isotope values reflects the ones of the diet using in silico simulation approaches. Authors showed (under very strong assumptions) that the answer is yes once considered the intra and inter variance of the isotopic baseline. The paper is well written, and i like very much the idea of testing the caveats of isotopic ecology using maths as already done by Flynn et al., (2018 Mar Biol), Jabot et al., (2017 Funct Ecol) or Ballutaud et al. (2022 Plos one) for instance. I cannot judge fully on the mathematics developped here (I am not mathematician and there is heavy stuff in this paper) but i can say with high certitude that a common isotope ecologist will be completely overwhelmed by the mathematical developments done in this paper. At least, i advice the authors to apply their findings to a case study as done by Yeakel et al. (2016). 

Authors: thank you for your helpful suggestions and edits on this paper. We particularly appreciate the mathematical nature of this makes it not an easy read for many / most ecologists and many of the co-authors would agree! Throughout the development of this paper, and in light of these and the editors comments we thought hard about including a case study, example or software to along with this paper. Ultimately we think that the best way forward is to let the mathematics and reasonably wide ranging simulation study here stand alone and for subsequent papers to build on this as a foundation. To do it justice to a real world application, we think there is a need for a substantial simulation study alongside real world data. We think that this requires a standalone study that can simply take the two key equations from this current paper as a starting point and explore the best approach to generating appropriate baseline corrections under a variety of real world situations where it is more than likely the case that the full pool of food resources is not available – else researchers could use mixing models to estimate directly diet proportions. There is also scope to develop some computational tools and introduce them into the existing R packages SIBER and/or nicheRover. This approach would benefit greatly from a numerical comparison of the various statistics currently employed to quantity isotope niche including ellipses, convex hulls and the methods we derive here. Again, we feel that the best way to do would be to use the maths exposition in our current paper as the foundation for this so as not to get bogged down trying to do it all at once. While we acknowledge that in its current form it probably is not as helpful to empirical ecologists as it might be with an example, we do feel that it is scientifically sound in its own right, and that it will be a key go-to paper for the foundations of the often ill-applied concept that isotopic niche is equal to trophic niche. 

Some more specific comments below. 

The submission suffers from the absence of line numbering that would have eased the review process 

Authors: apologies about this. We used the PLOSONE template for LaTeX exactly as provided and as instructed and without modification it did not generate line numbers. We have forced line numbers to appear in the current submission of the PDF, but have provided the TEX file without line numbers and a PDF built from the submitted TEX file. 

Intro 

• - First paragraph of the intro is about ecological niche while the reader would be much more interested in the trophic niche. Please clarify or explain. 

Authors: we intend to include the more general term niche here. Even though most researchers tend to focus on what the animal was eating, the same principle applies to understanding how an animal utilises space and how their isotope values might reflect sampling from food that varies isotopically over space and/or time. The conceptual for this relating to stable isotopes was established reasonably early on as stable isotope ecology took off as a discipline, e.g. Newsome et al 2007 A niche for isotopic ecology. We have added some text to the introduction to clarify and added a reference to Bearhop 2004 in the introduction. 

• - The statement « often difficult to observe organism’s niche » is not supported by any reference and is not clear at all. There is a bunch of literature on ecological niche. Please clarify and be specific regarding the trophic niche (stomach content may help and is observable in the field?) 

Authors: our use of “impossible” was something of a philosophical statement based on conversations about whether its even possible to define a niche in a truly quantitative way, and what we even mean by the word niche. But here is probably distracting. We have removed the word impossible, and left it as just “difficult” which is probably a fair and justifiable statement and is supported by papers cited in the introduction such as Bearhop 2004 and Newsome 2007. 

• - Second paragraph « Turnover rates which vary depending on the animal ». Authors means species or individual or both ? Also again possible confusion between ecological and trophic niche. Trophic niche is a sub set of the ecological niche ?

Authors: we have clarified that we mean both species and individual animal. Following from our previous response comment, we are comfortable with niche more broadly than just trophic niche. 

Model 

• - « Earlier than 1-5/r » is coming out from the blue. Please demonstrate. Ballutaud et al., (2022 plos one) estimated that the « 5/r » would be equal to 2*ln(2)/ln(r) 

Authors: this is something of a qualitative choice in this number. Indeed 5/r is very close to 2*ln(2)/ln(r) for most situations and both just mean that the exponential decay effect of the past on the present drops to almost negligible (which is the arbitrary decision) past this point. The link to Ballutaud is useful thanks and we make direct reference to this paper and alternative formulation in the text. 

• - « As long as diet temporal fluctuations are not too strongly correlated.... ». Could you provide a threshold in the correlation? Is it realistic in nature? This is an extremely important assumption which is not acknowledged anymore in the rest of the manuscript. 

Authors: this relates to the correlation between diet fluctuations and the fluctuations in the isotope concentrations in the food sources. These are more than likely independent in most situations, unless consumers are preferentially choosing food sources based on their isotopic signatures. There may be situations where seasonal variations could introduce some correlation of this type as an indirect effect, but for the most part this seems unlikely. We have added a short note to this effect.

• - Going from eq 3 to eq 4 is extremely harsh ! 

Authors: it probably is, but its actually a fairly straight forward implementation of commutativity of multiplication and addition coupled with the fact that with statistically independent variables in a dynamic system we can ignore their correlations. We have added some additional explanatory text to help guide the reader in this transition.

• - I suspect that an additional assumption must be made : All individual must have the same 

turnover rate. Otherwise another source of variation must be accounted for. If correct please add. If not please argue. 

Authors: good point. We have added this as a concept when r is introduced earlier in the model presentation but not complicated the maths further by expanding this additional variation. 

Result and discussion 

• - Page 9 « simplifying assumptions » please remind them + the one on turnover rate 

Authors: good point. Added. 

• - Phillips 2014 and Gorokhova 2018 are not in the reference list 

Authors: thanks. Added.

Conclusion 

- « We nonetheless expect to see a quantitative relationship » sounds like a weak conclusion. I prefer the « semi-quantative comparisons » terms. For example, Jabot et al. (2017 functional ecology) wrote « researchers should not hope to grasp subtle patterns of food web structure based solely on widely used isotopic indices. The remaining question is more about quantifying the thresholds (the limits as written by the authors) when variance in isotope values of the consumer does not reflect variance in the diet. No conclusions were drawn on these limits whereas this is a key word of the MS title. As a title i would prefer the sentence at the end of intro « how strong is the signal of diet .... » or better « How strengthen the signal of diet .... » 

Authors: agreed and changed.

---

## [Editor Report · Decision Letter 1]

26 Mar 2024

How strongly does diet variation explain variation in isotope values of animal consumers?

PONE-D-23-24341R1

Dear Dr. Jackson,

We’re pleased to inform you that your manuscript has been judged scientifically suitable for publication and will be formally accepted for publication once it meets all outstanding technical requirements.

Kind regards,

Jacob Freeman

Academic Editor

PLOS ONE

Additional Editor Comments (optional):

Thank you for your productive revisions. I believe that the paper meets Plos One criteria and is ready for publication.